# A Novel LED Light Radiation Approach Enhances Growth in Green and Albino Tea Varieties

**DOI:** 10.3390/plants12050988

**Published:** 2023-02-21

**Authors:** Xianchen Zhang, Keyang Chen, Ziyi Zhao, Siya Li, Yeyun Li

**Affiliations:** State Key Laboratory of Tea Plant Biology and Utilization, Anhui Agricultural University, Hefei 230036, China

**Keywords:** far-red light, photosynthetic, tea plants, growth, quality

## Abstract

Light, as an energy source, has been proven to strongly affect photosynthesis and, thus, can regulate the yield and quality of tea leaves (*Camellia sinensis* L.). However, few comprehensive studies have investigated the synergistic effects of light wavelengths on tea growth and development in green and albino varieties. Thus, the objective of this study was to investigate different ratios of red, blue and yellow light and their effects on tea plants’ growth and quality. In this study, Zhongcha108 (green variety) and Zhongbai4 (albino variety) were exposed to lights of different wavelengths for a photoperiod of 5 months under the following seven treatments: white light simulated from the solar spectrum, which served as the control, and L1 (red 75%, blue 15% and yellow 10%), L2 (red 60%, blue 30% and yellow 10%), L3 (red 45%, far-red light 15%, blue 30% and yellow 10%), L4 (red 55%, blue 25% and yellow 20%), L5 (red 45%, blue 45% and yellow 10%) and L6 (red 30%, blue 60% and yellow 10%), respectively. We examined how different ratios of red light, blue light and yellow light affected tea growth by investigating the photosynthesis response curve, chlorophyll content, leaf structure, growth parameters and quality. Our results showed that far-red light interacted with red, blue and yellow light (L3 treatments) and significantly promoted leaf photosynthesis by 48.51% in the green variety, Zhongcha108, compared with the control treatments, and the length of the new shoots, number of new leaves, internode length, new leaf area, new shoots biomass and leaf thickness increased by 70.43%, 32.64%, 25.97%, 15.61%, 76.39% and 13.30%, respectively. Additionally, the polyphenol in the green variety, Zhongcha108, was significantly increased by 15.6% compared to that of the plants subjected to the control treatment. In addition, for the albino variety Zhongbai4, the highest ratio of red light (L1 treatment) remarkably enhanced leaf photosynthesis by 50.48% compared with the plants under the control treatment, resulting in the greatest new shoot length, number of new leaves, internode length, new leaf area, new shoot biomass, leaf thickness and polyphenol in the albino variety, Zhongbai4, compared to those of the control treatments, which increased by 50.48%, 26.11%, 69.29%, 31.61%, 42.86% and 10.09%, respectively. Our study provided these new light modes to serve as a new agricultural method for the production of green and albino varieties.

## 1. Introduction

Tea (*Camellia sinensis* L.), an evergreen woody plant grown in diverse agroclimatic regions [1], is the world’s most economically important non-alcoholic beverage due to its enjoyable taste, various secondary metabolites and health benefits. The main secondary metabolites, theanine and tea polyphenols, contribute to the bitter, astringent, and sweet flavors that meet consumer demands across the world. In addition, secondary metabolites play an important role in health benefits such as the antioxidant and anti-inflammatory properties of tea and decreased risk of cardiovascular syndromes [2,3]. The tea yield and quality are largely determined by three factors: light, temperature and water, of which light is one of the most important environmental factors associated with plant growth and development [4].

Specific spectra exhibit different physiological responses. Red light (R) and blue light (B) are more effective as these wavelengths are absorbed by plants [5]. A growing body of evidence demonstrates that red light and blue light act on tea plants not only as an important energy source but also as an essential source of external signals. Transcriptomic and metabolomic analyses have shown that increased blue light intensity improved the lipid metabolism and flavonoid synthesis of tea shoots (one bud and two leaves) [6]. In addition, red light withering significantly promoted 10 free amino acids and theaflavin levels, and a sensory evaluation showed that the taste quality was significantly increased [7]. The contents of volatile fatty acid derivatives (VFADs), volatile phenylpropanoids/benzenoids (VPBs), and volatile terpenes (VTs) significantly increased in tea leaves (Jinxuan) exposed to red light [8]. However, monochromatic red or blue light could not meet the requirements for plant growth. For example, the leaf area and root biomass of cucumber were significantly impaired by monochromatic red light treatments [9]. Similar results showed that the Chla and Chlb contents were significantly decreased in Cordyline australis under 100% blue light treatment compared to 75% red with 25% blue light treatment [10].

In addition to visible light, the wavelength (λ > 700 nm) has long been considered to affect far-red light, and only long-wavelength far-red light makes a minimal contribution to photosynthesis. However, far-red light has synergistic activity with lights in the photosynthetically active radiation range (400–700 nm) [11]. Numerous studies have shown that adding far-red light at a shorter wavelength light significantly increases leaf photochemical efficiency and growth. Hwang et al. [12] found the supplemental far-red light significantly promoted the growth of tomato, red pepper, cucumber, gourd, watermelon and bottle gourd seedlings. Similar results demonstrated that adding far-red light immediately increased the quantum yield of photosystem II of lettuce under red/blue light [13]. Additionally, Ji et al. [14] reported adding far-red light increased greenhouse tomato yield (*Solanum lycopersicum*) and production. However, less attention has been paid to the effect of far-red light on tea plant growth.

As mentioned above, the effects of monochromatic light and far-red light on the different plants’ growth and quality of different plants have been reported. However, until recently, the synergistic effects of different wavelengths on tea plant growth and quality were largely overlooked for different tea varieties. Therefore, the effects of different red and blue ratios, together with far-red light, on the growth and quality of two varieties, Zhoncha108 (a green variety) and Zhongbai4 (an albino variety), were studied by investigating the photosynthetic rate, chlorophyll content, growth and quality content (free amino acids, theanine and tea polyphenol). The objective of the present study was to provide guidance on the design of light sources for tea plant cultivation in a controlled environment.

## 2. Results

### 2.1. Leaf Photosynthesis Performance

Here, the different types of light equipment (Figure 1A) and wavelengths (Figure 1B) are presented, and the ratios of red and blue light are shown in Figure 2A, Table 1 and Table 2.

Photosynthesis response curves were measured for fully expanded second leaves under each treatment. In the two tea varieties, Zhongcha108 and Zhongbai4, photosynthesis increased nonlinearly with the increasing light intensity (Figure 2B,E). With the addition of far-red light, the highest level of photosynthesis in Zhongcha108 was detected under the L3 treatment, which was significantly increased by 48.5% compared with that of the control treatments, ranked in the order of L6 < control < L5 < L4 < L1 < L2 < L3. In addition, the highest red/blue ratio (L1 treatment) led to the highest level of photosynthesis in Zhongbai4, and the level of photosynthesis was significantly higher by 44% compared with that of the control. The average photosynthesis ranked in the following order: L6 < L5 < control < L4 < L3 < L2 <L1. Similar trends in the light saturation point (Figure 2C,F) and light compensation point (Figure 2D,G) were observed in the two varieties.

### 2.2. Tea Plants Growth

The growth and morphology of the two tea varieties showed significant differences under different light treatments.

As shown in Figure 3, the growth of Zhongcha108 was the greatest in the plants grown under an RB ratio of 1.5 with the addition of far-red light. The greatest new shoot length, length of the internode, number of newly expanded leaves, new leaf area and new shoot weight, were observed under the L3 treatment, being significantly increased by 70.43%, 25.97%, 32.63%, 15.60% and 76.38% compared to the control treatments. For Zhongbai4, the higher ratio of red light significantly promoted tea growth. Thus, the maximum new shoot lengths, length of the internode, number of newly expanded leaves, leaf area and new shoot weight were observed under the L1 treatment. In contrast, the lowest ratio of red light (the highest of blue light) significantly impaired tea growth, and the length of the internode, the number of new leaves and new shoot weight were significantly decreased by 14.52%, 13.31% and 14.29% compared to the control treatments (Table 3).

### 2.3. Leaf Structure and Chlorophyll Concentration

To further clarify the role of light in tea growth, the leaves’ microscopic structure was observed, and paraffin sections were made. The morphology of the second tea leaves showed significant differences under different treatments (Figure 4).

A consistent trend in the tea plants was also evident, whereby the thickest spongy tissue and palisade tissue were observed under the L3 treatment in Zhongcha108 was significantly increased by 30.45% and 24.75% compared to the control. Additionally, the thickness of the leaves was the lowest when the plants were grown under the higher ratio of blue light treatment. The thickness of the spongy tissue and palisade tissue under the L5 and L6 treatments showed approximately a 5.04% reduction in comparison with that of the control treatment. For the albino variety, the greatest thickness of the leaves was observed in Zhongbai4 grown under the highest ratio of red light. The L1 treatment significantly improved the thickness of the spongy tissue and palisade tissue by 15.46% and 29.48% in Zhongbai4 compared to that of the control treatments, while the lowest thickness was observed in the plants grown under the higher ratio of blue light treatment (Table 4).

Consistent with its growth, Zhongcha108 grown under 60%R + 30%B with far-red light also showed increased Chla (56.33%) and Chlb (85.42%) compared with the control treatments. In the case of Zhongbai4, the plants had the highest chlorophyll contents with 75%R + 15%B, which were significantly increased by 35.37% and 60.47% compared to the control, and the lowest red light treatment led to the lowest chlorophyll content (Table 5).

### 2.4. Tea Quality

To further explore the effects of different light treatments on the tea quality, the total free amino acids, theanine and tea polyphenol were measured, respectively. Among the different treatments of Zhongcha108, R/B = 1.5 combined with far-red light (L3 treatments) resulted in the highest level of tea polyphenol for Zhongcha108 (Figure 5A). Additionally, the levels of C (catechin, 50.0%), EC (epicatechin, 21.00%), GC (gallocatechin, 61.94%), EGC (epigallocatechin, 13.64%), ECG (epicatechin gallate, 21.02%), GCG (gallocatechin gallate, 39.29%) and EGCG (epigallocatechin gallate, 13.18%) significantly increased compared with those of the control treatments (Table 6). However, the tea leaves exhibited the lowest levels of total free amino acids (Figure 5B) and theanine (Figure 5C). Therefore, the addition of far-red light resulted in the highest ratios of tea polyphenols and free amino acids (Figure 5D).

Similarly, LED light supplying R/B = 5 (L1 treatments) increased the total tea polyphenol in the Zhongbai4 leaves as compared to the other treatments. The highest ratio of red light treatment significantly increased this parameter by 14.9% compared with that of the control treatments (Figure 5A). An increase in the accumulation of C (39.23%), EC (24.7%), GC (78.4%), EGC (22.5%), ECG (35.8%), GCG (26.0%) and EGCG (9.9%) was observed (Table 6). The total free amino acid (Figure 5B) and theanine (Figure 5C) contents under the L1 treatment decreased by 33.04% and 38.32%, respectively, compared with those of the control treatments. Thus, the lower level of free amino acid and higher polyphenol resulted in a higher ratio of tea polyphenol and free amino acid (Figure 5D).

## 3. Discussion

### 3.1. Far-Red Light Promoted Photosynthesis and Growth in Zhongcha108

Light quality is an important component that is strongly associated with tea plant growth and development [6]. A previous study reported that visible light (400–700 nm) was the most effective wavelength for photosynthesis in tea plants. For example, blue light (peak at 460 nm) could promote the accumulation of anthocyanins and catechins in Zhonghuang3 [15]. Recently, Ye et al. [16] reported that red light increased the contents of catechins and flavonol glycosides in fresh tea leaves. However, the effect of far-red light on photosynthesis has long been considered inefficient, offering a minimal contribution [17]. Interestingly, a growing body of evidence has shown that the synergistic effect between far-red light and shorter wavelengths is beneficial for the enhancement of plant growth and developmental processes [18]. Red light combined with far-red light significantly increased photosynthetic CO_2_ assimilation, thus resulting in a higher net photosynthetic rate, total biomass, leaf area and chlorophyll content in soybean seedlings [19]. Similar results showed that with the addition of far-red light, two lettuce cultivars (Yanzhi and Red Butter) exhibited a higher root and shoot weight compared with samples subjected to white light treatment [20]. However, only a few studies on the effects of far-red light on the photosynthetic parameters, leaf structure and growth of tea plants have been reported.

In our study, which aimed to quantify the effect of far-red light on the photosynthetic efficiency of tea plants, we found that adding far-red light significantly increased the photosynthetic rate in Zhongcha108 (Figure 2). Thus, greater increases in the leaf area and plant height were observed after five months of exposure to far-red light (Table 2). It may be that far-red facilitated the PSII reaction centers and stimulated them to use the absorbed photons more efficiently, a process that was associated with an increase in net photosynthesis [17]. The latest study by Lanoue et al. [21] showed that far-red light treatments resulted in a higher efficiency of PSII in the fifth leaf of peppers (*Capsicum annuum*), which enhanced the height, leaf area and internode length. Zou et al. [22] found that adding FR light facilitated a higher plant radiation use efficiency (RUE) in lettuce (Lactuca sativa L. cv. ‘Tiberius’) and improved the leaf area and fresh weight. Similar results demonstrated that the photosynthetic photon flux density and total dry weight of lettuce plants (Lactuca sativa cv. Expertise RZ) were increased with the addition of far-red light at three planting densities compared with the plants under red light treatment alone [23].

In addition, a recent study demonstrated that far-red light could regulate the endogenous hormone levels of plants to stimulate hypocotyl elongation and the extensibility of the plant cell walls, thus resulting in an increased plant height and leaf area [24]. Gibberellins, auxin and ethylene are directly involved in regulating far-red light-mediated shoot growth changes [25]. Islam et al. [26] reported that an increased far-red (FR) proportion in natural light at the end of the day enhanced the leaf area and shoot elongation in Poinsettia (*Euphorbia pulcherrima*) through the promotion of total GA metabolism. Additionally, Kurepin et al. [27] found that the addition of far-red light addition appeared to initiate a reduction in the ethylene levels of 7-day-old sunflower seedlings (*Helianthus annuus* L.), promoting an increase in the hypocotyl length and stem elongation.

Therefore, the far-red light-promoted growth of Zhongcha108 may facilitate the PSII reaction centers or regulate hormone metabolism. However, the molecular mechanism remains poorly understood; thus, more work will be carried out in our future studies.

### 3.2. High Ratio of Red Light Promoted Photosynthesis and Growth in an Albino Variety, Zhongbai4

It is well known that plant growth is strongly associated with photosynthesis [28]. Photosynthetic pigments are the material basis of photosynthesis, and their composition and content directly affect the photosynthetic rate of leaves. Approximately 70% of chloroplasts are located in palisade cells, and the remainder are predominantly located in spongy mesophyll cells [29]. In our study, the highest ratio of red light (Li treatment) induced the leaves of zhongbai4 to grow thicker (Figure 4), and the greatest thickness for palisade tissue and spongy tissue was detected in the plants subjected to the L1 treatment (Table 4). Therefore, among the seven treatments, the highest chlorophyll levels observed in the L1 treatment group significantly increased photosynthesis (Figure 2E), which promoted tea growth in regard to the leaf area, new shoot length, new shoot biomass and internode length (Table 3). Similarly, the increasing proportion of red light significantly increased the shoot growth, mainly through its effects on the thickness of the palisade mesophyll, spongy mesophyll tissue and photosynthetic rate [30]. Pennisi et al. [31] reported that LED light supplying R/B = 3 or 4 increased the total chlorophyll content and plant growth of sweet Basil as compared to R/B = 1.

In contrast, the highest ratio of blue light treatments (L5 and L6 treatments) impaired the thickness of the leaves (Figure 4) and chlorophyll content (Table 5), resulting in lower photosynthetic efficiency and degree of tea plant growth (Figure 2E). As mentioned above, we speculated that the poorly developed chloroplasts might be one explanation for the lower photochemical conversion efficiency, which had a negative effect on seedling growth [32]. Spaninks et al. [33] found that blue light reduced the shoot growth and development of Arabidopsis thaliana and tomato (Solanum lycopersicum). However, blue light has been reported to enhance single-leaf photosynthetic efficiency in lettuce [34]. Li et al. [35] also found that blue light promoted the development of leaf tissue structures and chloroplasts in apple plants; however, it impaired the apple plant’s height and root growth. The above-mentioned study indicated that there are species-specific differences. Therefore, more work will be carried out in our future studies. In summary, the R/B = 5 treatment is an appropriate mode for the albino variety, Zhongbai4 (Table 2).

### 3.3. Prospects for Tea Agricultural Application

Light-emitting diodes (LEDs) have been adopted in a wide range of applications in controlled agricultural facilities in recent years, as they provide opportunities to fine-tune the light ratio and manipulate crop growth [36]. A growing body of research has reported that the application of light-emitting diodes (LEDs) has been conducted to further understand its effect on plant growth. A recent study demonstrated that blue photo-selective shade netting increased the chlorophyll content and photosynthetic efficiency of Vanilla planifolia [37]. In our study, the L1 (75% red, 25% blue, 10% yellow) treatment significantly improved the growth of the albino variety, Zhongbai 4, while far-red light addition (L3 treatment) remarkably promoted photosynthesis, resulting in a higher leaf area and shoot biomass in Zhongcha108.

In ornamental horticulture, the commercial value of the plant mainly depends on the quality [10]. For tea plants, theanine is an important component associated with tea quality [38]. In our study, the L3 and L1 treatments significantly promoted the growth of Zhongcha108 and Zhongbai4, respectively, but impaired theanine synthesis (Figure 5). Interestingly, among the seven treatments, the highest blue light ratios (L6 treatments) resulted in the maximum theanine levels in Zhongcha108 and Zhongbai4, respectively. We suggest that providing far-red light and a high ratio of red light benefits tea growth in Zhong108 and Zhongbai4 in the seedling stage, while a high ratio of blue light improves the tea quality in the mature period. Therefore, our results show that these light modes have the potential to serve as new agricultural modes for green and albino varieties in the future.

## 4. Materials and Methods

### 4.1. Plant Material and Experimental Settings

The green variety, Zhongcha108, and albino variety, Zhongbai4 (1-year seedlings, approximately 16–18 cm in height), were obtained from tea seedling breeding bases in Nanjing city, Jiangsu Province. The tea plants were carefully washed and fixed in a plastic pot with a radius of 12 cm, which was filled with peat, perlite and vermiculite (1:1:1). Then, the tea plants were transferred to an artificial climate chamber. The setting conditions were as follows: photoperiod and relative humidity of 12 h·d^−1^ and 40%, respectively. The tea plants were grown for 2 weeks before light treatments. The LEDs were equipped with light plates (Wuhan Doublehelix Biology Science and Technology Co., Ltd.). All the plants were subjected to 200 µmol·m^−2^·s^−1^ irradiance. The different light treatments with different red/blue light ratios included the control, L1, L2, L3, L4, L5 and L6, as shown in Table 1 and Table 2. In addition, the control treatments (white light) were simulated from the solar spectrum. The tea plants were plated in a climate chamber (6×4 m and located at 31°52′ N 117°15′ E.) on 26 May 2021 and harvested after 5 months.

### 4.2. Chemicals and Instruments

The chemicals were as follows: 95% ethanol, calcium carbonate, quartz sand, stannous chloride sodium carbonate, folinol, gallic acid, potassium dihydrogen phosphate, ninhydrin, sulfuric acid, ascorbic acid, L-theanine, acetic acid, methanol and acetonitrile. The instruments comprised the following: a fiber optic spectrometer, spectral analyzer, portable plant photosynthesis apparatus, ultraviolet-visible spectrophotometer, electronic analytical balance, vacuum freeze dryer, drying oven, centrifuge and high-performance liquid chromatography.

### 4.3. Determination of Photosynthetic Characteristics

Photosynthetic response curves of the fully expanded second leaves of nine plants from each treatment group were monitored with a portable photosynthetic instrument (CIRAS-3, PP Inc., USA) [39]. The light response curves were measured using different light intensities in the range from 0 to 2000 µmol·m^−2^·s^−1^. The starting light intensity was 200 µmol·m^−2^·s^−1^, followed by 200, 100, 50, 20, 0, 300, 400, 500, 600, 800, 1000, 1200, 1400, 1600, 1800 and 2000 μmol·m^−^^2^·s^−^^1^. The order of measurements was arranged randomly for each repetition, and each tea plant measurement was repeated three times. Data were recorded when the photosynthesis rate maintained a steady state at every light intensity level. In addition, the light saturation point and light compensation point were analyzed using Ye’s model [40].

### 4.4. Chlorophyll Concentration

Samples were collected from the first and second leaves of ten plants for each treatment. Leaves were weighed 0.2 g (fresh weight, FW). The tea leaves were ground with 10 mL of 80% acetone, the homogenate was filtered, and then the samples were centrifuged at 4000×*g* for 10 min to collect the supernatant. The optical density was measured with a U-5100 spectrophotometer (Hitachi, Japan) at 663 nm (OD663) and at 645 nm (OD645) for chlorophyll a (Chl a) and chlorophyll b (Chl b), respectively, using the following formulae: Chla concentrations = 12.72 × A663 − 2.59 × A645, Chlb concentrations = 22.88 × A649 − 4.67×A663.
Chlorophyll concentration (mg/g) = Chl concentrations × V/Fw (1)
where V and FW indicate the volume of the reaction system and the weight of the isolated tea leaves, respectively [41].

### 4.5. Leaf Structure

After 5 months, the second leaves of the tea plants were collected from each treatment group for microscopic analysis. Paraffin sections were obtained as described by Zheng et al. [10]. The thickness of the spongy tissue and palisade tissue was measured using Imagine-Pro Plus software (Media Cybernetics company, Rockville, MD, USA), respectively.

### 4.6. Determination of Tea Plant Growth

For the growth analysis, measurements of twelve plants were taken for each treatment when new buds had ceased to sprout. The new shoot length, number of newly expanded leaves and length of the internode were measured. The new shoot weight was precisely detected using an electronic balance (Mettler Toledo, Shanghai, China). The leaf area was measured using an area meter (LI-3100C, LI-COR Biosciences, Lincoln, NE, USA).

### 4.7. Free Amino Acids, Theanine and Tea Polyphenol

The total free amino acids were extracted in boiling water and analyzed by the ninhydrin coloration staining method [5]. The levels of theanine and tea polyphenol were measured according to our previous methods with minor modifications. The theanine contents were analyzed by HPLC (Thermo Fisher UltiMate 3000, Waltham, MA, USA) [3]. The total polyphenols were extracted with 70% (*v*/*v*) methanol at 70 °C, and the content was analyzed according to the Folin–Ciocalteu method, as described by Chang et al. [2] In addition, the catechins were extracted and measured for high-performance liquid chromatography (HPLC) analysis, as previously described by Lin et al. [42].

### 4.8. Statistical Analysis

All measurements were based on different replicate plants. All data analyses were performed using SPSS 26.0 software. Data are presented as the mean ± SD. The statistical analysis included a 1-way analysis of variance (ANOVA), and significant differences between the means were tested using Duncan’s multiple range test at 95% confidence.

## 5. Conclusions

Comparing the different light modes and tea varieties, far-red light combined with shorter wavelengths (L3 treatments) was beneficial for the growth of Zhongcha108, while the highest red light ratio (L1 treatments) was an appropriate mode for the albino variety, Zhongbai4. In addition, we suggest that providing a high ratio of blue light improves theanine levels in the mature period.

## Figures and Tables

**Figure 1 plants-12-00988-f001:**
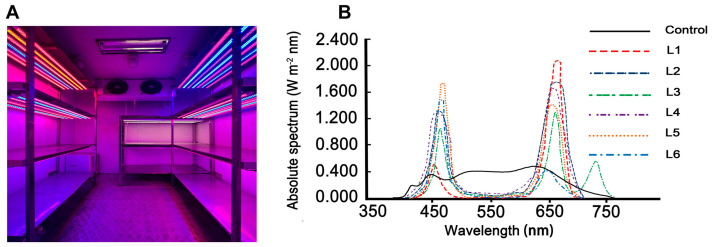
Spectral distribution of different light treatments. (**A**) All types of light equipment in an artificial climate chamber, (**B**) Spectral composition in different light treatments.

**Figure 2 plants-12-00988-f002:**
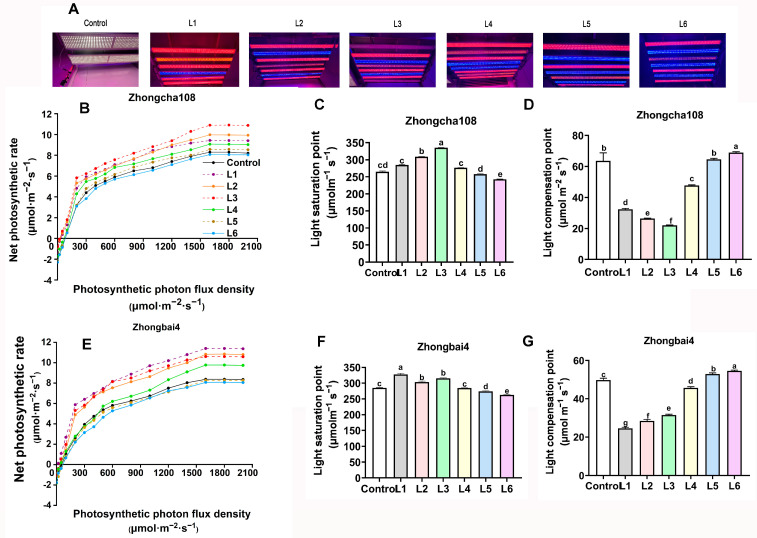
The effect of different light treatments on photosynthesis in two varieties. (**A**)Different light equipment; (**B**,**E**)The effect of different light treatments on photosynthesis at multiple light intensities in Zhongcha108 and Zhongbai4, respectively; (**C**,**F**) The effect of different light treatments on light saturation points in Zhongcha108 and Zhongbai4, respectively; (**D**,**G**) The effect of different light treatments on light compensation points in Zhongcha108 and Zhongbai4, respectively. Note: Different lowercase letters on the top of the bar chart indicate significant differences at the level of *p* < 0.05.

**Figure 3 plants-12-00988-f003:**
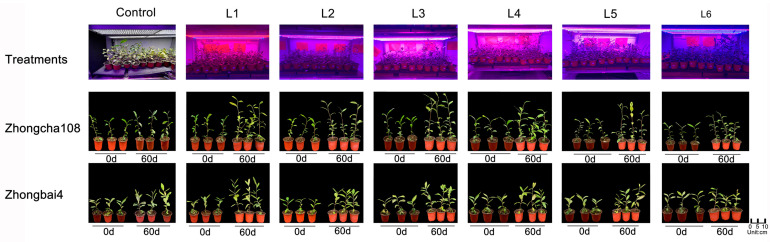
The effect of different light treatments on the phenotype in two varieties.

**Figure 4 plants-12-00988-f004:**
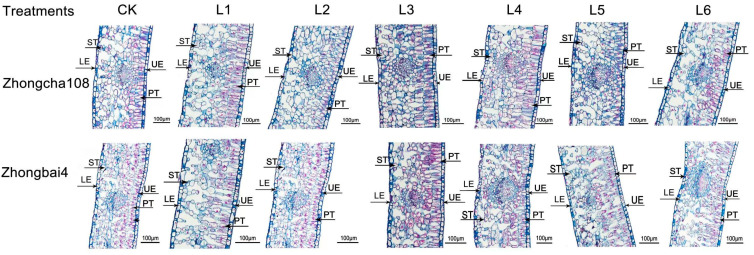
The effect of different light treatments on the leaf structure in two varieties. Note: ST, PT, UE and LE indicated spongy tissue, palisade tissue, upper epidermal cells and lower epidermal cells.

**Figure 5 plants-12-00988-f005:**
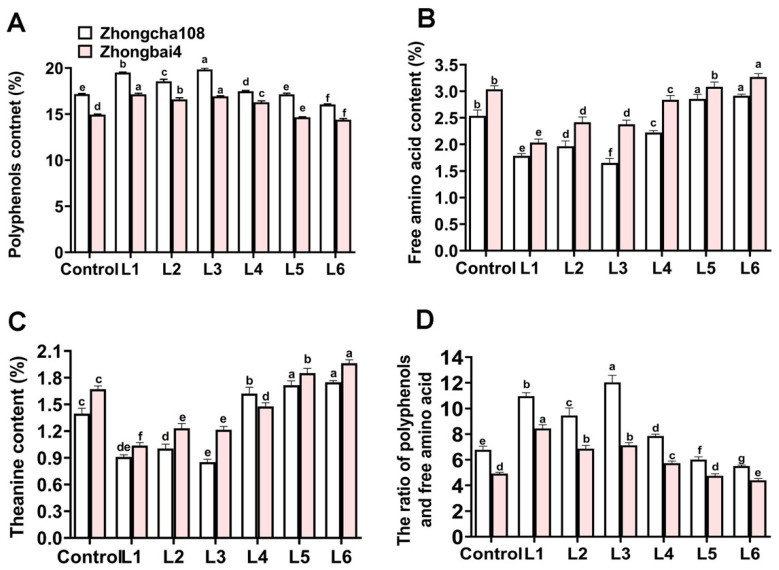
The effect of different light treatments on the quality in two varieties. (**A**) The effect of different light treatments on polyphenol content, (**B**) The effect of different light treatments on free amino acids, (**C**) The effect of different light treatments on theanine level, (**D**) The effect of different light treatments on the ratio of polyphenol and theanine. Note: Different lowercase letters on the top of the bar chart indicate significant differences at the level of *p* < 0.05.

**Table 1 plants-12-00988-t001:** Different light ratios of red, blue and yellow light of seven treatments.

Light Treatment	Control	L1	L2	L3	L4	L5	L6
Red light	19.6%	75%	60%	45%	55%	45%	30%
Blue light	5.1%	15%	30%	30%	25%	45%	60%
Yellow light	75.3%	10%	10%	10%	20%	10%	10%
Far Red light	0%	0%	0%	15%	0%	0%	0%

**Table 2 plants-12-00988-t002:** The parameters of light quality.

	Number of Tubes	Red Light	Blue Light	Yellow Light	Far-Red Light	Peak Wave Length (nm)	Main Wavelength (nm)	Spectral Half Width (HW/nm)	Light Intensity (μmol^−1^·m^−2^·s^−1^)
Light Treatments									
Control		White light				626.0	577.4	253.6	202–257
L1		4	1	1	0	657.0	541.9	23.4	200–253
L2		4	2	1	0	658.0	564.5	23.8	195–261.4
L3		3	2	1	2	659.0	562.0	24.2	200–248
L4		4	2	1	0	458.0	458.0	22.7	201–258.7
L5		3	3	1	0	458.0	448.3	22.8	201–235
L6		3	3	1	0	458.0	458.0	23.6	202–252

**Table 3 plants-12-00988-t003:** The effects of different light qualities on shoot growth in Zhongcha108 and Zhongbai4. Means followed by different letters differ significantly by Duncan (*p* < 0.05).

Light Treatments	Tea Varieties	Control	L1	L2	L3	L4	L5	L6
New shoot length (cm)	Zhongcha108	13.87 ± 3.79 bc	15.58 ± 6.29 bc	16.65 ± 5.26 b	23.64 ± 4.28 **a**	18.34 ± 4.52 b	16.08 ± 5.65 bc	11.24 ± 2.61 c
	Zhongbai4	8.26 ± 2.79 ^b^	12.43 ± 2.72 ^a^	9.30 ± 2.43 ^ab^	10.72 ± 3.26 ^ab^	9.12 ± 3.98 ^ab^	9.53 ± 4.01 ^ab^	8.16 ± 4.80 ^b^
Internode length (cm)	Zhongcha108	4.89 ± 1.26 b	5.12 ± 1.28 ab	5.42 ± 1.71 ab	6.16 ± 0.62 **a**	5.29 ± 1.00 ab	4.66 ± 1.28 b	4.38 ± 1.30 b
	Zhongbai4	2.41 ± 0.80 ^b^	4.08 ± 0.92 ^a^	3.02 ± 0.99 ^ab^	3.06 ± 0.87 ^ab^	2.60 ± 1.11 ^b^	2.68 ± 1.69 ^b^	2.06 ± 1.28 ^b^
The number of new leaves	Zhongcha108	3.83 ± 0.94 b	3.75 ± 1.06 b	4.17 ± 0.72 b	5.08 ± 0.79 **a**	4.40 ± 0.98 ab	4.33 ± 0.88 ab	3.58 ± 0.79 b
	Zhongbai4	3.08 ± 0.79 ^cd^	4.83 ± 1.34 ^a^	3.50 ± 1.00 ^bcd^	4.75 ± 1.22 ^a^	4.17 ± 1.27 ^abc^	4.58 ± 1.08 ^ab^	2.67 ± 0.65 ^d^
New leaf area (cm^2^)	Zhongcha108	17.17 ± 2.16 ab	19.09 ± 3.45 a	18.91 ± 4.02 a	19.85 ± 2.78 **a**	17.40 ± 3.92 ab	17.60 ± 3.81 ab	15.23 ± 2.48 b
	Zhongbai4	15.44 ± 2.24 ^b^	20.32 ± 4.45 ^a^	19.95. ± 4.42 ^a^	16.82 ± 2.94 ^ab^	19.16 ± 4.39 ^a^	17.43 ± 3.15 ^ab^	15.21 ± 2.69 ^b^
New shoot biomass (g)	Zhongcha108	0.72 ± 0.15 cd	0.75 ± 0.11 cd	0.89 ± 0.16 bc	1.27 ± 0.28 **a**	1.03 ± 0.50 ab	0.82 ± 0.15 bcd	0.59 ± 0.04 d
	Zhongbai4	0.63 ± 0.16 ^cd^	0.90 ± 0.18 ^a^	0.66 ± 0.06 ^c^	0.81 ± 0.09 ^ab^	0.67 ± 0.07 ^c^	0.77 ± 0.08 ^b^	0.54 ± 0.04 ^d^

**Table 4 plants-12-00988-t004:** The effect of different light qualities on the thickness of mesophyll tissue in Zhongcha108 and Zhongbai4. Means followed by different letters differ significantly by Duncan (*p* < 0.05).

Light Treatments	Chlorophyll A (mg/g FW)		Chlorophyll B (mg/g FW)		Chlorophyll (A + B) (mg/g FW)	
	Zhongcha108	Zhongbai4	Zhongcha108	Zhongbai4	Zhongcha108	Zhongbai4
Control	1.42 ± 0.04 cd	0.82 ± 0.02 ef	0.48 ± 0.01 d	0.43 ± 0.02 d	1.90 ± 0.04 cd	1.25 ± 0.02 d
L_1_	1.74 ± 0.01 b	1.11 ± 0.016 ^a^	0.84 ± 0.01 a	0.69 ± 0.01 ^a^	2.57 ± 0.09 b	1.80 ± 0.03 a
L_2_	1.57 ± 0.06 bc	0.861 ± 0.01 d	0.53 ± 0.04 cd	0.55 ± 0.02 c	2.10 ± 0.01 cd	1.41 ± 0.02 d
L_3_	2.22 ± 0.06 ^a^	1.02 ± 0.02 b	0.89 ± 0.02 ^a^	0.62 ± 0.02 b	3.11 ± 0.05 a	1.64 ± 0.03 b
L_4_	1.48 ± 0.27 cd	0.91 ± 0.02 c	0.73 ± 0.07 b	0.61 ± 0.02 b	2.20 ± 0.09 c	1.52 ± 0.01 c
L_5_	1.23 ± 0.01 de	0.83 ± 0.01 e	0.67 ± 0.03 b	0.43 ± 0.03 de	1.90 ± 0.03 de	1.26 ± 0.03 de
L_6_	1.10 ± 0.01 e	0.79 ± 0.02 ^f^	0.55 ± 0.02 c	0.400 ± 0.01 e	1.65 ± 0.03 e	1.19 ± 0.03 e

**Table 5 plants-12-00988-t005:** The effects of different light treatments on chlorophyll content in Zhongcha108 and Zhongbai4. Means followed by different letters differ significantly by Duncan (*p* < 0.05).

Light Treatments	Sponge Tissue Thickness (μm)		Palisade Tissue Thickness (μm)		New Leaf Thickness(μm)	
	Zhongcha108	Zhongbai4	Zhongcha108	Zhongbai4	Zhongcha108	Zhongbai4
Control	141.73 ± 10.01 ^c^	137.98 ± 7.49 ^ab^	101.49 ± 4.65 ^c^	96.82 ± 6.77 ^cd^	315.19 ± 7.90 ^cd^	284.23 ± 8.14 ^bc^
L_1_	149.56 ± 11.70 ^bc^	159.31 ± 8.96 ^a^	109.21 ± 6.59 ^c^	125.35 ± 5.91 ^a^	323.43 ± 10.2 ^cd^	312.91 ± 9.9 ^a^
L_2_	166.85 ± 9.84 ^ab^	141.12 ± 11.04 ^ab^	109.73 ± 4.15 ^bc^	107.05 ± 6.80 ^bcd^	331.33 ± 13.77 ^bc^	291.34 ± 12.96 ^abc^
L_3_	184.88 ± 7.76 ^a^	145.04 ± 12.80 ^ab^	126.61 ± 9.82 ^a^	113.34 ± 7.11 ^abc^	357.14 ± 12.20 ^a^	297.86 ± 16.21 ^ab^
L_4_	177.64 ± 10.60 ^a^	155.24 ± 15.59 ^a^	119.74 ± 10.80 ^ab^	119.45 ± 8.87 ^ab^	348.94 ± 11.82 ^ab^	301.41 ± 9.16 ^ab^
L_5_	134.58 ± 5.78 ^c^	126.33 ± 7.54 ^b^	96.70 ± 6.80 ^c^	96.54 ± 12.67 ^cd^	300.69 ± 7.95 ^d^	272.90 ± 13.71 ^c^
L_6_	136.09 ± 14.02 ^c^	132.84 ± 8.79 ^b^	100.18 ± 11.90 ^c^	92.97 ± 9.37 ^d^	310.76 ± 20.88 ^cd^	273.60 ± 9.45 ^c^

**Table 6 plants-12-00988-t006:** The effect of light treatment on the catechinic acid in Zhongcha108 and Zhongbai4. Means followed by different letters differ significantly by Duncan (*p* < 0.05).

Light Treatments	Tea Varieties	Control	L1	L2	L3	L4	L5	L6
C (mg/g)	Zhongcha108	5.34 ± 0.13 ^de^	7.37 ± 0.15 ^b^	6.17 ± 0.12 ^c^	8.01 ± 0.02 a	5.62 ± 0.08 ^d^	5.30 ± 0.25 ^e^	4.53 ± 0.16 f
	Zhongbai4	4.97 ± 0.08 ^d^	6.92 ± 0.17 ^a^	5.84 ± 0.14 ^c^	6.50 ± 0.11 ^b^	5.72 ± 0.22 ^c^	4.67 ± 0.21 ^e^	4.28 ± 0.09 ^f^
EC (mg/g)	Zhongcha108	12.33 ± 0.10 ^cd^	14.80 ± 0.11 ^a^	12.63 ± 0.24 ^c^	14.92 ± 0.39 a	13.15 ± 0.29 ^b^	12.27 ± 0.27 ^cd^	11.99 ± 0.33 d
	Zhongbai4	9.03 ± 0.22 ^c^	11.26 ± 0.20 ^a^	10.32 ± 0.17 ^b^	10.76 ± 0.75 ^ab^	8.93 ± 0.21 ^c^	8.87 ± 0.24 ^c^	8.29 ± 0.49 ^c^
GC (mg/g)	Zhongcha108	2.68 ± 0.23 ^c^	4.29 ± 0.05 ^a^	3.01 ± 0.04 ^b^	4.34 ± 0.11 a	3.17 ± 0.13 ^b^	3.02 ± 0.29 ^b^	2.17 ± 0.08 d
	Zhongbai4	2.13 ± 0.09 ^cd^	3.80 ± 0.15 ^a^	2.42 ± 0.13 ^b^	3.74 ± 0.16 ^a^	2.28 ± 0.05 ^bc^	1.90 ± 0.13 ^e^	1.70 ± 0.12 ^e^
EGC (mg/g)	Zhongcha108	13.64 ± 0.28 ^c^	15.58 ± 0.20 ^a^	15.08 ± 0.37 ^b^	15.91 ± 0.41 a	13.85 ± 0.21 ^c^	13.52 ± 0.34 ^c^	13.10 ± 0.18 d
	Zhongbai4	11.39 ± 0.20 ^cd^	13.95 ± 0.62 ^a^	12.17 ± 0.38 ^bc^	13.11 ± 0.70 ^ab^	12.13 ± 0.82 ^bc^	11.40 ± 0.20 ^cd^	10.56 ± 0.48 ^d^
ECG (mg/g)	Zhongcha108	1.95 ± 0.22 ^bc^	2.26 ± 0.15 ^ab^	2.21 ± 0.13 ^b^	2.36 ± 0.05 a	2.02 ± 0.14 ^c^	1.98 ± 0.11 ^cd^	1.84 ± 0.09 d
	Zhongbai4	1.59 ± 0.24 ^cd^	2.16 ± 0.80 ^ab^	1.74 ± 0.07 ^c^	2.01 ± 0.05 ^a^	1.63 ± 0.17 ^c^	1.48 ± 0.19 ^d^	1.36 ± 0.22 ^d^
GCG (mg/g)	Zhongcha108	3.08 ± 0.07 ^cd^	3.68 ± 0.10 ^b^	3.27 ± 0.18 ^c^	4.29 ± 0.27 a	3.22 ± 0.15 ^c^	3.06 ± 0.24 ^cd^	2.66 ± 0.29 d
	Zhongbai4	2.85 ± 0.20 ^bc^	3.59 ± 0.12 ^a^	3.28 ± 0.16 ^ab^	3.42 ± 0.14 ^a^	3.19 ± 0.18 ^ab^	2.63 ± 0.39 ^c^	2.48 ± 0.45 ^c^
EGCG (mg/g)	Zhongcha108	109.54 ± 1.17 ^cd^	122.47 ± 1.91 ^a^	112.25 ± 1.96 ^b^	123.98 ± 1.62 a	110.91 ± 1.96 ^bc^	109.01 ± 1.18 ^cd^	106.91 ± 0.93 d
	Zhongbai4	95.05 ± 1.59 ^d^	104.45 ± 1.30 ^a^	99.34 ± 1.05 ^c^	101.68 ± 1.12 ^b^	94.56 ± 1.30 ^d^	92.36 ± 1.87 ^de^	90.33 ± 1.59 ^e^

## Data Availability

The data presented in this study are available on request from the corresponding author. The data are not publicly available due to privacy.

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
