# Peer review of "A Novel LED Light Radiation Approach Enhances Growth in Green and Albino Tea Varieties"

_plants, 2023, doi:10.3390/plants12050988_

Round 1

Reviewer 1 Report

The research titled “A novel approach of LED light radiation enhances growth in green and albino tea varieties” is a fantastic work for providing information to tea growers. I have some minor suggestions to improve the paper for the publication.

Abstract

Write results little bit elaborately in abstract.

Line 11, replace plant to leaves.

Line 13, add light after blue.

Line 16, L2 (red 60%, blue 15% and yellow 10%) is not 100%, is it okay?

Introduction

Line 30, make tea scientific name italic. Also, check all scientific name through the MS.

Line 37-39, rewrite the sentence “A growing body ………. external signal”.

Line 41, use Wang et al. [7], and use similar format through the MS

Line 52, add a space after 700.

Line 58, make scientific name italic, add a space before significantly and use full stop at the end of the sentence.

Line 62, use citation properly, correct m2 to m2.

There is no linkage between ending part of the introduction. Suddenly you start The objective of the present study was to characterize the ……………

Results

Explain Figure 1B. Why you use Figure 2A? There is nothing in the written content.

In case of table 1, for Control treatment, check the ratio.

Figure 2B and 2E, use high resolution figure.

In Figure 2 legend, mention statistical information. Also check for all table and figure legend.

Table 2, looking so odd, improve the arrangement of the table, use proper alignment.

Figure 4, use bigger size photo so that the structure can clearly see.

Line 142-143, use full form of C, EC, GC, ………………

Line 178, replace maximumly to significantly.

Line 267, Check it.

Line 268, add space after respectively.

Line 272, add space after Wang et al.

Line 286-289, Check them. Add full stop, spacing where necessary.

In all table, you use value ± values and letters, but there is no mention anywhere in the paper about the value after ±, is it SE or SD, also what does it mean by different letters (Both in tables and figures).

Reviewer 2 Report

This manuscript is A novel approach of LED light radiation enhances growth in 2 green and albino tea varieties.

I do not think this manuscript is a highly scientifically certain-level paper. However, at the current level, it is good information to know the effects of LED light ratio on plant growth, catechinic acid, and photosynthesis of tea plants (Zhongcha108 and Zhongbai4).

- I think this study lacks more adequate results and cannot meet the requirements published in Plants Journal.
- There are a lot of typing and grammatical errors in the manuscript. And a thorough grammar and English language revision by a native English scientific expert is required.
Abstract: The abstract must be made crisp. Some details of the experimental method and results are to be included. The experimental results must also indicate the increase/decrease over control values. The level of significance is to be indicated.

Introduction:

line 32 The specific role of each of the secondary metabolites in tea and human health benefit must be highlighted in brief. need to detail it.

 Is there any information on the levels of secondary metabolites determined from tea? If so then please add it.

The objective of the study is not written properly; the language must be improved so as to reflect its importance.

Materials and methods:

The source of seeds and the plant cultivar must be added. Moreover, the source of all chemicals, solvents, instruments, etc is to be mentioned. Site details (including geographical coordinates) must be mentioned in brief. What is the size of the artificial climate chamber?

Results and discussion

The logic is not clear enough and the focus is not prominent enough.

It is recommended to refine the results and describe the important results in a specific way. In addition, try not to always use the same sentence pattern in the presentation of the results, so as to make the article read more fluently and clearly.

The discussion section is very weak
Conclusion
This section needs to be more precise and include some recommendations also.
In view of the above-mentioned reasons, the ms cannot be accepted in its present form.

More detail needs to check in the attached file.

Author Response

Pleaes see the attachment.

Reviewer 3 Report

Manuscript “A novel approach of LED light radiation enhances growth in green and albino tea varieties“, by Zhang et al., covers one of the most interesting questions of plant biology investigated both by fundamental and applied approaches. These studies are providing answer to neglected aspect of previous experimental studies which did not consider response differences induced by different spectral composition of white light sources.

            The advance of LED technology which replaced the incandescent and neon lamps enabled effects of single monochromatic light and their combinations to be studied over a variety of physiological processes. Although many good reports appeared in the last decade, the interest for studies dealing with spectral composition of light is still very strong, specially in the group of economically interesting crops including tea (Camellia sinensis L.).

            Zhang et al., manuscript offers a wide and detailed study of various plant parameters measured in one year old, potted plants grown in controlled conditions for a whole season (5 months). The amount of manual work that was spent in conducting of this experiment was very high compared to currently popular, molecular techniques.  In general my opinion on this study on tea plants cultured in growth chambers is positive, but text needs much work to be done by authors on improvements, plus a detailed proofreading.

            Manuscript text is rather short, as many points especially in the introduction are not well explained. Problem is the very poor English language which makes text difficult to understand in many places. Detailed proofreading is required, but even then authors should expand their sentences adding more details, descriptions and explanations for the benefit of readers.

            In the introduction description of expectations how blue and red light will affect the growth of tea plants is mostly missing. Effects are announced but not explained. There are statements that these lights exert certain effects with a list of other recent studies which were dealing with the red and blue light in tea plants. These report need to be presented with more details, specially the one written in Chinese (7).

            After reading the text several times I am still not sure about the manufacturer, and other specifications of employed monochromatic light sources.

            Authors need to declare what is the role of yellow light in L1 – L6 treatments, providing also its wavelength.  In my opinion yellow light will somewhat support the red light.

            Far-red light is a mystery, readers will not understand how, when and in what quantities were provided to plants as it is not mentioned in Table 1.          

            For control white light we have its spectral composition but there is no mention of other data.

            Authors should avoid using word „photons“, since it complicates the story, stay with light of certain wavelength.

            Readers (me too) would certainly like to know more what Zhogbai4 looks like, where is the defect (in which tissues and organs) and what pigments are affected.

            Tables 2 and 5 and perhaps some others too, would appear better if columns and rows change places. Thus data for L1 to L6 should be presented in columns and not in rows. In this case table will have 7 instead of 11 columns and it would resemble Table 1.

            Your non-Chinese speaking reviewer was not very happy to find that nearly 15% of listed references were  of papers written in Chinese only, placing them out of his reach. Many readers will have similar problem and develop a similar attitude.

            Similarly, Chinese authors are more than favored in the reference list. Is it some kind of a new trend?  I am aware that most papers dedicated to light effects in plants in the last decade were produced by Chinese authors. Especially work of Deng XW focused on COP9 and function of signalosome in plant cells. But lack of non-Chinese scientist is certainly a disadvantage here in this paper, which is easy to demonstrate. Strong and specific effect of blue light on growth of potato shoots was demonstrated a long time ago by Seabrook JEA 2005, and I believe this finding was later used to cover plants under cultivation with colored sheaths to modify their growth. Kurepin et al 2007  have a fantastic early far red/red ratio finding demonstrating that this ratio significantly affects production of cytokinins in potato. I would also recommend papers by Spanikis et al 2020 and Islam et al.  2014 – for the sake of balance.  I added some more papers in a list at the end which may be of interest to authors, without any obligations.

List    

Seabrook JEA (2005) Light effects on the growth and morphogenesis of potato (Solanum tuberosum) in vitro: a review. Am J Potato Res 82:353–367

Kurepin LV, Walton LJ, Reid DM (2007) Interaction of red to far red light ratio and ethylene in regulating stem elongation of Helianthus annuus. Plant Growth Regul 51:53-61. https://doi.org/10.1007/s10725-006-9147-x

Islam MA et al (2014) Impact of end-of-day red and far-red light on plant morphology and hormone physiology of poinsetia. Scientia Horticulturae 174: 77-86

Spanikis Kiki et al. (2020) Regulation of Early Plant Development by Red and Blue Light: A Comparative Analysis Between Arabidopsis thaliana and Solanum lycopersicum. Front. Plant Sci., 23 December 2020 Sec. Crop and Product Physiology https://doi.org/10.3389/fpls.2020.599982   

Some interesting references

Yao  Xu-yang, LiuXiao-ying, Xu Zhi-gang, Jiao Xue-le (2017) Effects of light intensity on leaf microstructure and growth of rape seedlings cultivated under a combination of red and blue LEDs.  Journal of Integrative Agriculture 2017, 16(1): 97–105.  https://doi.org/10.1016/S2095-3119(16)61393-X 

Li et al. BMC  Effects of red and blue light on leaf anatomy, CO2 assimilation and the photosynthetic electron transport capacity of sweet pepper (Capsicum annuum L.) seedlings Plant Biology (2020) 20:318 https://doi.org/10.1186/s12870-020-02523-z

Jin Wenqing et al. (2021). Adding Far-Red to Red-Blue Light-Emitting Diode Light Promotes Yield of Lettuce at Different Planting Densities. Front. Plant Sci., 15 January 2021, Sec. Crop and Product Physiology, https://doi.org/10.3389/fpls.2020.609977  

Qin NX, Xu DQ, Li JG, Deng XW (2020) COP9 signalosome: Discovery, conservation, activity, and function. J Integr Plant Biol 62:90-103. https://doi.org/10.1111/jipb.12903

JingY, Lin R (2020) Transcriptional regulatory network of the light signaling pathways. New Phytol. 227:683-697. https://doi.org/10.1111/nph.16602    

Xu F, He SB, Zhang JG, Mao ZL, Wang WX, Li T, Hua J, Du S, Xu PB, Li L, Lian HL, Yanh H-Q (2018) Photoactivated CRY1 and phyB interact directly with AUX/IAA proteins to inhibit auxin signaling in Arabidopsis. Mol Plant 11:523-541. https://doi.org/10.1016/j.molp.2017.12.003

Round 2

Reviewer 1 Report

The improvement of the paper is satisfiable. 

Reviewer 2 Report

This manuscript is A novel approach of LED light radiation enhances growth in 2 green and albino tea varieties.

Plants journal is a high-quality journal with IF: 4.658 and Q1 in the category 'Plant Sciences. Therefore, there are some errors in the manuscript which is not acceptable.

1. All tables in this manuscript do not have footnotes about the significant differences among treatments.

2. I do not know which soft program was used to analyze data in research.

3. There are no significance and regression lines in the tables.

4. line 11 deletes "yield and quality"

Reviewer 3 Report

I seldom encounter authors who were ready to perform such a fast and thorough improvement of their manuscript. It makes reviewer proud that he contributed, increasing the manuscript quality on the benefit of readers and the scientific community. Manuscript in the present form is acceptable for publishing and I wish authors all the best in their future studies.

Round 3

Reviewer 2 Report

This manuscript can be accepted in its present form.